# Analysis of attacking styles and goal-scoring in the 2021/22 Women's Super League

**Lizzie Craven**[1], **Patrick Oxenham**[2], **Jayamini Ranaweera**[3] *

**1** Faculty of Sport, Technology and Health Sciences, St Mary's University, Twickenham, London, England, **2** Fulham Football Club, Motspur Park, London, England, **3** Department of Sport and Physical Activity, Edge Hill University, Ormskirk, England

☯ These authors contributed equally to this work.
* jayamini.ranaweera@edgehill.ac.uk

**Data Availability Statement:** Data relevant to this paper are available from OSF at https://doi.org/10.17605/OSF.IO/RQUVD.

**Funding:** The author(s) received no specific funding for this work.

## Abstract

The rapid rise in elite women's football has increased the demand for female specific research to enable more accurate and appropriate assessments of tactical performance. Thereby, this study aimed to describe goal scoring in relation to different attacking styles during a Women's Super League (WSL) season. Specifically, 1179 attacking sequences leading to shots on target performed by all 12 teams in the 2021/22 WSL season were analysed. The style of attack for each attacking sequence was characterised by research guided key performance indicators and recorded with the outcome of the subsequent shot on target. Descriptive results indicated that most shots (27.23%) were originating from combinative organised attacks, while set plays accounted for the most goals (27.08%), with fast organised attacks demonstrating the best goal conversation rates (53.33%). Outcomes of a chi-square test highlighted a significant (but weak) association between attacking styles and shot outcomes ($\chi_4^2 = 9.87$, $P = 0.043$) in the considered WSL season, with shots originating from set plays resulting in significantly more goals than expected (AR = 2.45). Overall, the results can be useful for practitioners when formulating tactical game plans and training sessions, while also providing multiple opportunities for future research in tactical analysis of women's football.

## 1. Introduction

Women's football has been recognised as the fastest growing sport in the world, with the quality, investment and interest experiencing dramatic exponential growth [1]. Such rising interest and engagement in women's football have been notable in both domestic and international competitions such as the Women's Super League (WSL), the Women's FA Cup, the European Championships and the FIFA World Cup [2, 3]. Increased fan interaction and a subsequent rise in funding has enabled clubs to invest more heavily into supporting coaches and players [4]. This has facilitated greater access to performance analysis in the female game and greater levels of integration of the discipline within women's teams [5]. Research illustrates that the implementation of performance analysis within day-to-day practice has been directly linked to

**Competing interests:** The authors have declared that no competing interests exist.

maximising the growth of women's football [6]. Subsequently, leading to a comparable increase in performance analysis research within women's football [7]. However, the extent in which this can be applied to women's football heavily depends on the social-contextual environment such as the limited female-specific data and financial resources available to clubs [8]. The level of support and overall professionalism varies largely across elite women's football, with working conditions for coaches, players and support staff being seriously inadequate in some cases [9]. This will negatively impact the extent in which analysis is implemented and the subsequent level of support analysts can provide towards both the coaching process and player development [10]. Consequently, providing research that can be directly applied into the elite women's game will help support both coaches and analysts.

Analysing tactical aspect of a team's performance has been noted as fundamental in providing an extensive and accurate assessment and enhancement of performance [11]. Tactical analysis considers a variety of key performance indicators that are likely to result in successful performance and winning games [12]. Previous male-focused research examined individual elements of a team's tactical performance from attacking and defensive perspectives. Attacking variables considered as vital for successful performance within this research include ball possession [13], passing variables [14] and a variety of shooting variables [15]. From a defensive perspective, variables such as ball recovery [16], defensive pressure [17] and defensive balance [18] have been considered in outlining the tactical trends in a team's performance. This has allowed analysts to quantify the patterns and trends that constitute successful performances by individuals and teams [19].

At the centre of recent male-focused tactical research has shifted towards analysing playing styles instead of individual performance indicators to produce more effective summaries of tactical performance [20]. Styles of play are defined as the "tactical behaviours" employed by a team that aims to "achieve the attacking and defensive objectives in the game" [21]. It has been noted that assessing a team's attacking proficiency aids in forming an accurate assessment of that team's tactical set up and extent of performance success [22]. Interestingly, the low-scoring, invasive nature of football leads to attacking performance success often being centred around the frequency of goals scored and how this impacts wins achieved [23]. While scoring a goal is recognised as the most successful and influential action during a game, it should be considered that only 1% of team possessions lead to a goal being scored [24]. It is therefore vital to transfer the focus of tactical analysis from the simplistic view of goal frequencies to a broadened overview of the construction of goal-scoring attacks [15]. As research suggests, the factors that make up the construction of attacking play determine a team's tactical style and are directly linked to the frequency and success of goal-scoring [25].

Hughes and Franks [24] categorised a team's tactical attacking style as "direct" or "possession" and this categorisation has been subsequently integrated in contemporary literature [15, 23, 26]. A "direct" attacking style is characterised as using a limited number of long and or vertical passes to transition the ball quickly to the opposing goal [15]. In contrast, a "possession" attacking style utilises a greater number of passes over a longer duration to progress the ball towards the opposition goal [27]. While this helps to distinguish between the two opposing strategies, research has evolved and attacking styles are now categorised with greater granularity. These studies differentiated between "counterattacks" produced in offensive transitions [28], "direct attacks" characterised by long passes [29], "fast attacks" that use quick and vertical passes [30] and "combinative attacks", characterised by long ball possessions [31]. However, the actual implementation of each team's style of play not only depends on the team's preparation but it is strongly influenced by the interaction with the opponent's tactical behaviour, as well as by the contextual variables.

To ensure contextual variables are considered, Aranda et al. [32] designed the 'REOFUT' framework—a research-guided observational tool that aims to categorise a team's style of play. The 'REOFUT' framework outlines the combination of variables and actions that exist in the start, development, penultimate and final stages of an attack, all of which determine a team's attacking style. Research has considered the variables that formulate the start of an attacking sequence, including type of start [33], starting zone [34] and initial behaviour from both the attacking and defensive team [35]. Similarly, variables that make up the development of a team's offensive play have been evaluated in the REOFUT' framework. These include key performance metrics such as: the number of passes [36], type of passes [37], pass penetration [38], duration [39] and possession width [40]. Finally, both the penultimate and final action have also received attention in the research, with variables such as assist zone [25], assist action [31], type of goal-scoring action [36] and goal-scoring zone [25] being scrutinised. The combination of previously mentioned attacking variables determines the type of attack performed by a team during offensive phases of play and has been linked to variations in the frequency and success in shots on goal taken [25, 34]. A prevalent finding in previous research is that attacking performance is heavily focused on men's football and fails to consider the variations that exist within female football [41, 42], and as a result could limit the relevance when applying conclusions in analyst practice in the women's game [42]. Practitioners and analysts working within women's football are often required to consult and utilise the findings from male-focused research due to the lack of gender-specific research that exists, particularly from a tactical perspective [41]. While research surrounding men's football has successfully transitioned towards analysing attacking performance from a broadened, stylistic, tactical perspective, female-specific research often reverts to analysing simplistic individualised attacking variables [43]. Given the previously discussed significance of goal scoring and attacking patterns of play, it is crucial for practitioners operating in women's football to build upon existing evidence to gain a more comprehensive understanding of tactical performance. In this context, expanding and deepening applied research on women's football is essential in extracting the tactical insights needed by practitioners to execute strategic decision-making processes [44], while also helping to reduce the existing disparity in tactical analysis compared to the men's game. Therefore, as an initial step to help bridge those gaps in current literature, this study aims to describe goal scoring based on different attacking styles in the Women's Super League (WSL) tournament. Specifically, the study initially examines literature to identify and characterise different attacking styles and associated variables relating to women's football. Next, guided by those findings, we provide a descriptive account of goal scoring that had occurred based on the attacking styles in the considered season. Finally, we examine the potential relationship between the style of attack and outcome of the resulting shot on target during the considered WSL season.

## 2. Methodology

### 2.1. Operational definitions

The key performance indicators (KPIs) and operational definitions used for this study were established in three stages. Initially, a literature review was conducted (by the lead author LC) to identify KPIs and operational definitions associated with attacking styles of play. Guided by the previously stated 'REOFUT theoretical framework', the style of each attacking sequence was determined based on a combination of key performance indicators [32]. The REOFUT framework outlines and details the key action variables associated with five main attack types: combinative organised attack, fast organised attack, direct organised attack, counterattack and set plays and is used in attacking research [15, 23, 31]. Action variables presented in the start

**Table 1. Categorisation of attack styles from the selected KPIs and relevant action variables.**

| Key Performance Indicators | | | | | | | | Type of Attack |
|---|---|---|---|---|---|---|---|---|
| Type of start | Defensive organisation | Type of passes | No. of passes | % of penetrative passes | Starting zone of attack | Width of attack | Duration of attack | |
| Regain / Restart | Balanced | Short / Mixed | 4–6 / 7+ | Low % of Penetrative / Medium % of Penetrative | Defensive / Central / Attacking | Low / High | 9–15 seconds / 16 + seconds | Combinative organised |
| Regain / Restart | Balanced | Long | 0–3 | High % of Penetrative | Defensive / Central | Low | 0–8 seconds | Direct organised |
| Regain / Restart | Balanced | Short / Mixed | 0–3 | High % of Penetrative | Defensive / Central / Attacking | High | 0–8 seconds | Fast organised |
| Regain | Balanced / Unbalanced | Short / Mixed | 0–3 / 4–6 | High % of Penetrative | Defensive / Central / Attacking | Low | 0–8 seconds | Counter attack |
| Set play | NA | Short / Medium / Long | NA | NA | Central / Attacking | NA | NA | Set play |

Note: see supplementary materials (S1 Table) for the relevant evidence from literature used to develop the action variable conditions pertaining to each attack style.

and development phases of an attacking sequences, as outlined by the 'REOFUT theoretical framework' assist in determining the type of attacking style. A breakdown of each attacking style and associated action variables are presented in Table 1. Importantly, specific evidence from the literature that were used to guide the categorisation of the 5 attack styles from the action variables specified in Table 1 are provided in the supplementary materials (S1 Table). Moreover, since 'set plays' are characteristically different to open play attacking styles, characteristics and associated action variables are defined by a limited range of literature (see S1 Table). Moreover, if the attacking sequence and associated variables differed from the guided combination of variables from the 'REOFUT theoretical framework' and related research, the attack was labelled as a 'nonconforming attack'. Thereafter, while each action variable was drawn from the start and development phases of an attack by the 'REOFUT theoretical framework', a further literature review was conducted (by the lead author LC) to determine the validity of these variables and provide a more comprehensive overview. Each variable was further divided into categorical groups as guided by research and presented in Table 2. Following the coding of each attacking sequence, the outcome of the resulting shot on target was noted either as 'Goal' or 'No Goal' as guided by research (definitions presented in Table 3) [45]. Once the KPIs were identified and operational definitions were validated through a review of the existing literature, they were presented to a panel of experts. The panel further validated the definitions, KPIs, action variables, and the categorisation of attack styles, confirming their relevance and accuracy. The expert panel was made up of football coaches (n = 3) who had a combined experience of 24 years working in women's football. During each individual interview (n = 3), KPIs were presented to each coach firstly by outlining the definition (and visual schematic where relevant) of each KPI and the associated action variables. To further strengthen the validation, selected video footage pertaining to the attack sequences were also used to guide engagement with the coaches. Specifically, each video clip was played twice for a coach (or more if requested) whereby, final agreement was reached.

## 2.2. Data collection

In terms of the overall study, this investigation utilised a descriptive design to describe the attack styles that led to a shot on target by the twelve teams that competed in an English Women's Super League (WSL) season. Subsequently, all 132 games played in the 2021/2022 WSL

**Table 2. Operational definitions of the selected key performance indicators and action variables.**

| Category | Description | Action variable | Operational definition |
|---|---|---|---|
| Type of start | The way in which a team starts a period of possession, considering whether the ball was in or out of play | Regain | When a player/team gains possession of the ball during any open play sequence excluding receiving the ball from a player of the same team. |
| | | Restart | When a player/team starts an attacking sequence that results from a restart of the game: 1) From any area of the pitch. 2) The attacking team is tactically unprepared to shoot at goal (players do not change positions). 3) The attacking team aims to keep possession of the ball from a sequence of passes. This includes goal-kicks, kick-offs, throw-ins and free-kicks won and taken in a team's own half. |
| | | Set play | When a player/team starts an attacking sequence that results from a set play: 1) Won and taken in the attacking half. 2) The attacking team is tactically prepared to shoot at goal. 3) The attacking team aims to pass/cross the ball and generate a shot in one or two passes. This includes all corner kicks, penalty kicks and free kicks taken in the attacking half. |
| Defensive organisation | The position of the opposition's defensive players in the first 3 seconds of an attacking sequence. | Balanced | The opposition has 3 or more players, excluding the goalkeeper, protecting the space between the ball and the goal at each moment of attempting to stop the progress of the attack and regaining possession of the ball. |
| | | Unbalanced | The opposition has less than 3 players, excluding the goalkeeper, protecting the space between the ball and the goal at each moment of attempting to stop the progress of the attack and regaining possession of the ball. |
| Type of passes | The average distance of passes played in an attacking sequence. | No Pass | No pass was completed prior to a shot being taken during an attacking sequence. |
| | | Short | The majority of completed passes travel over a distance less than 30 metres. |
| | | Mixed | There was a combination of both short and long passes during an attacking sequence. |
| | | Long | The majority of completed passes travel over a distance 30 metres or more. |
| Number of passes | The number of passes completed prior to a shot on goal during an attacking sequence. | 0–3 | 0–3 passes completed prior to the goal being awarded. |
| | | 4–6 | 4–6 passes completed prior to the goal being awarded. |
| | | 7+ | 7 or more passes completed prior to the goal being awarded. |
| % of penetrative passes | The number of passes that travel past an opposition player(s) in relation to the total number of passes during team possession, noted as a percentage. | Low penetrating possession (0–33%) | The number of penetrating passes equate to 0–33% of total passes completed during an attacking sequence. |
| | | Medium penetrating possession (34–66%) | The number of penetrating passes equate to 34–66% of total passes completed during an attacking sequence. |
| | | High penetrating possession (67–100%) | The number of penetrating passes equate to 67–100% of total passes completed during an attacking sequence. |
| Starting zone of attack (see Fig 1) | The zone of the pitch from where the possession originated. | Defensive | A third of the pitch estimated from the goal line to the middle zone. |
| | | Central | A third of the pitch estimated from the end of the defensive zone to the start of the attacking zone. |
| | | Attacking | A third of the pitch estimated from the end of the middle zone to the opponent's goal line. |
| Width of attack (see Fig 2) | The number of longitudinal lanes in which possession passes through by the attacking team during an attacking sequence. | Narrow attack (1–2 lanes) | During possession, the ball moves through 1 or 2 of the longitudinal lanes during an attacking sequence. |
| | | Wide attack (3–4 lanes) | During possession, the ball moves through 3 or 4 of the longitudinal lanes during an attacking sequence. |

*(Continued)*

**Table 2.** (Continued)

| Category | Description | Action variable | Operational definition |
|---|---|---|---|
| Duration of attack | The time taken (seconds) from the start of the attack to the shot being taken. | 0–8 (seconds) | The time taken (seconds) from the start of the attack to the shot being taken was between 0–8 seconds. |
| | | 9–15 (seconds) | The time taken (seconds) from the start of the attack to the shot being taken was between 9–15 seconds. |
| | | 16+ (seconds) | The time taken (seconds) from the start of the attack to the shot being taken was over 16 seconds. |

season were represented in the sample with 1179 attacking sequences being obtained from those games. Clips of each attacking sequence of the 132 games played in the 2021/2022 WSL from full game footage was obtained from match analysis software Wyscout (Chiavari, Italy). In relation to postgraduate dissertation ethics approval criteria, the aforementioned data collection process (and overall research project) was approved by the Faculty of Sport, Technology and Health Sciences, St Mary's University ethics committee. Moreover, written consent was obtained from the video data provider prior to notational analysis. As specified below, guided by the definition provided by Pollard and Reep [46] each clip began at the start of a team possession. All types of attack from Table 1 utilised this definition. "A team possession commences when a player gains ball possession in a controlled and deliberate manner by any means excluding receiving the ball from a player from the same team. This action may be followed by an individual action such as dribbling and carrying the ball or a series of passes between one team. The possession ends when one of the following occurs: 1) a goal is awarded; 2) the ball goes out of play; 3) the ball touches an opposition player to change the sequence of play (a tackle, interception or save). Brief touches that fail to alter the direction of play or sequence are excluded."

The attacking sequence was judged to be complete when one of the following occurred: 1) a goal had been awarded by the referee; 2) the ball travelled out of play (save or block) [47, 48]. During the analysis of each attacking sequence, a coding system was developed to guide the process of recording the notational data. This was done to ensure consistency when coding each clip by the same individual. The consequent data were logged in an observational instrument, designed and formatted in Microsoft Excel Version 16.0.67 (Microsoft Cooperation, United States) allowing for the filtration and processing of data for further analysis. To ensure consistency when coding each attacking sequence, operational definitions and visual schematics were used as guides (see Tables 2 and 3; Figs 1 and 2). Following coding of each attacking sequence, the outcome of the resulting shot was noted either as 'Goal' or 'No Goal' as guided by research (definitions presented in Table 3) [45].

**Table 3. Operational definitions for the outcome of the resulting shot on target.**

| Category | Action variable | Operational definition | Notes |
|---|---|---|---|
| Shot on target | Goal | The ball passing the goal line inside the dimensions of the goalpost resulting in the referee awarding a goal. | **Includes:** Any ball contact that results in the ball passing over the goal line within the goal post dimensions during the attacking sequence. For example: deflections, secondary contacts or goalkeeper saves that result in a goal being awarded. |
| | No Goal (Shot on Target) | Any attempt on goal by the attacking team that travelled or would have travelled in between the goal post dimensions and cross the goal line. | **Includes:** defensive blocks or a goalkeeper save. |
| | | | **Excludes:** Any attempt that directly contacted the crossbar or goal post. |

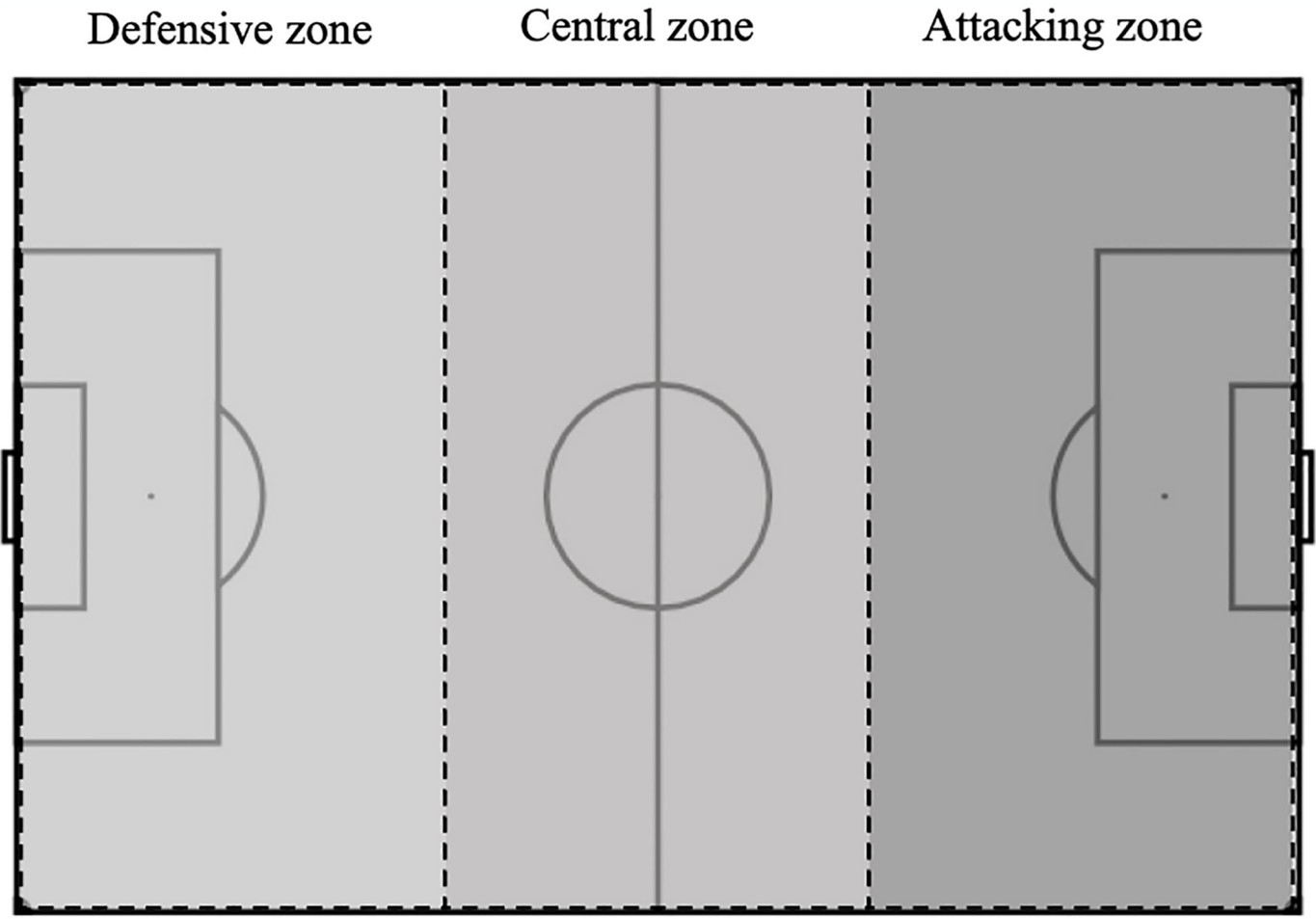

**Fig 1. Visual schematic of the three zones from which attacks start.**

## 2.3. Reliability

As the primary data collection instrument, the intra-rater reliability of the lead researcher when performing the notational analysis was done by coding a random sample of 118 attacks (10% of the total) on two separate occasions. Importantly, the ratings rounds were separated by a 1-week washout period [49]. To quantify the intra-rater reliability, Cohen's Kappa (k) statistic was calculated for: type of attack (k = 0.96), type of start (k = 0.96), opposition defensive balance (k = 0.95), type of passes (k = 0.95), number of passes (k = 1), percentage of penetrative passes (k = 0.95), starting zone of attack (k = 0.96), width of attack (k = 0.97) and duration of attack (k = 0.99), resulting in an overall Kappa value of 0.97. Guided by literature, agreement was interpreted as slight (0.01–0.20), fair (0.21–0.40), moderate (0.41–0.60), substantial (0.61–0.80) and almost perfect (0.81–0.99) [50, 51]. Thereby, the overall notation analysis illustrated almost perfect intra-reliability, indicating that quality of the generated data was suitable for the subsequent analysis.

## 2.4. Statistical analysis

**2.4.1. Descriptive and inferential statistics.**   Descriptive analysis of the sample was executed by calculating the relative frequencies and percentages for each style of attack. In terms

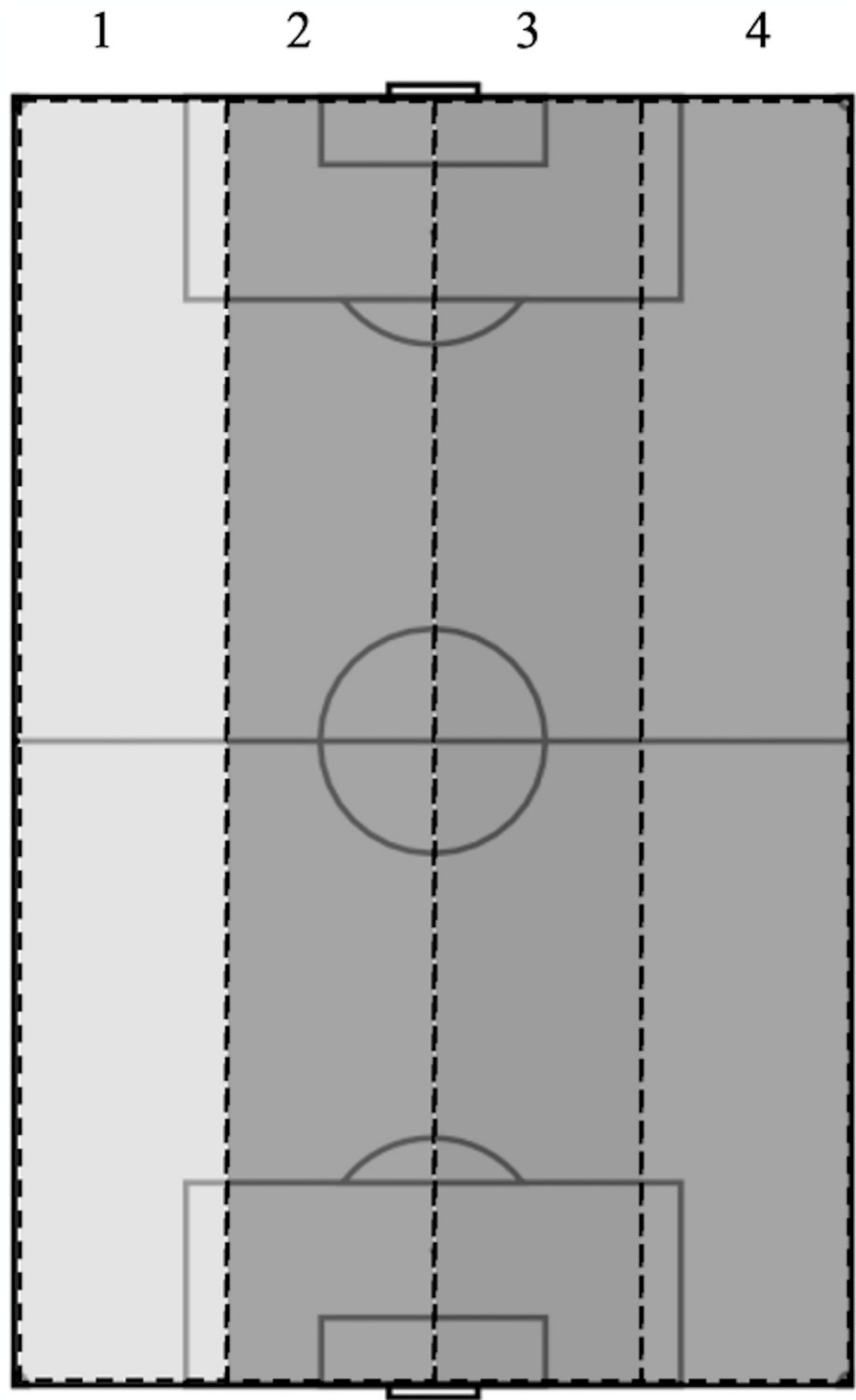

**Fig 2. Visual schematic of the four longitudinal lanes representing the pitch width.**

of the inferential analysis, the potential relationship between style of attack and outcome of the resulting shot on target of the 12 teams that competed in the 2021/22 WSL season was analysed using a chi-square test of independence, with an alpha level of 0.05 considered as the threshold for statistical significance. Importantly, based on the degrees of freedom of the test, effect size of the chi-square test was determined using Cramer's V and interpreted using the thresholds specified by Cohen [52]. Moreover, as a measure of post-hoc analysis, adjusted standardised residuals (AR) were used to quantify the relationships between each attacking style and shot outcome. Specifically, in accordance with the norms and guided by similar recent research [53], magnitude of the residuals was assessed in relation to the criterion ± 2.0 (1.96), where AR values above or below the thresholds were deemed to signify significantly greater or lower observed counts than the expected frequencies. Finally, all data analysis reported in this study was conducted using R programming language within the RStudio integrated development environment (IDE).

**2.4.2. Sample sizes.** To guide sample size requirements, an a-priori sample size calculation was conducted using G*power 3.1.9.2. Subsequently, at a pre specified alpha level 0.05, 80% power and with 4 degrees of freedom, the calculation illustrated that a minimum 133 shots on target/attacking sequences were required to detect a medium effect size of 0.3 [54]. Therefore, since the initial sample of shots on target considered in the study (n = 1179) was much larger than that specified by the priori calculation, the expected power for the analysis was achieved. Thereby, minimising the formulation of potential Type II errors.

## 3. Results

From the 132 WSL games that were played in the 2021/22 season, 1179 attacking sequences that led to a shot on target were analysed. Open plays accounted for 77.95% of the total shots on target (n = 919), whereas the remaining 22.05% were from 'set plays' (n = 260). As presented in Fig 3, considering all shots pertaining to each style of attack, most shots on target (n = 321) were from combinative organised attacks (27.23%), whereas fast organised attacks (n = 15) accounted for the lowest proportion (1.27%) of shots on target. Interestingly, set plays (22.05%) and counterattacks (20.87%) had resulted in approximately similar proportions of shots on target. Moreover, 21.46% of the shots (n = 253) considered in the sample did not meet the criteria of the 5 attack types and were categorised as 'nonconforming'. However, as reported in the immediate paragraphs, for transparency and consistency, those 253 shots were considered when describing the sample, but were omitted when conducting and reporting outcomes of the inferential analysis.

In terms of goals scored, 32.57% (384) of the total attacking sequences (n = 1179) resulted in a goal. Specifically, as presented in Fig 4, 27.08% (n = 104) of all those goals were scored from set plays, with combinative organised attacks also contributing to 26.04% (n = 100) of goals in the 2021/2022 WSL season. Interestingly, as expected due to the lowest number of shots on target, the least proportion of goals were created from fast organised attacks, contributing only to 2.08% (n = 8) of the total goals scored in the season.

When considering conversation rates of the shots on target, as highlighted in Fig 5, although contributing to the lowest percentage of goals scored in the season, 53.33% of the shots on target from fast organised attacks resulted in goals. Similarly, among the attack styles contributing to larger proportions of the total shots, set plays demonstrated the best conversation rates with 40% of those shots leading to goals. However, while most attack sequences had adopted combinative organised attacks to create scoring chances, the results indicate that only 31.15% of those sequences yielded positive outcomes (i.e., goals). Additionally, the results also

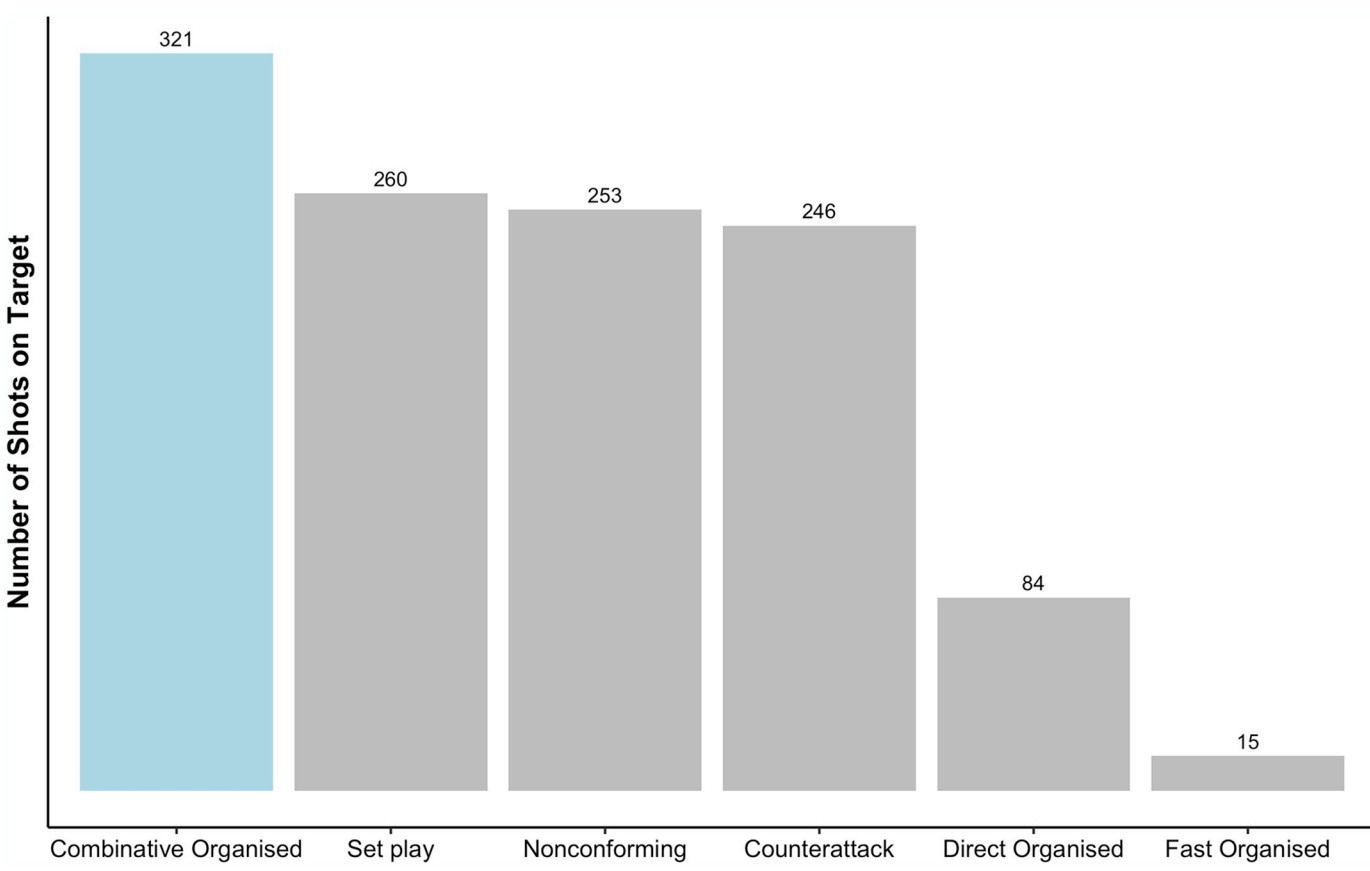

**Fig 3. Total shots on target based on the different types of attacks in the 2021/2022 WSL season.**

indicate that direct organised attacks had the least conversation rate (27.38%) of shots on target resulting in goals during the 2021/2022 WSL season.

Regarding inferential statistics, guided by relevant literature [21, 41], as specified previously, the 253 'nonconforming' shots categorised in the sample were excluded for the hypothesis testing, resulting in 926 shots on target being considered for the final analysis. Thereby, outcomes of the chi-square test demonstrated a significant association between the style of attack and outcome of the shot on target, $\chi^2(4) = 9.87$, p = 0.043; V = 0.1. However, the small effect size indicated that the observed association between the variables was weak. Moreover, based on the adjusted standardised residuals (AR), the results further suggested that the observed number of goals scored was more than expected for set plays (AR = 2.45). Interestingly, none of the other residuals yielded significant outcomes, suggesting minimal contributions to the association observed between the style of attack and outcome of the shot on target.

## 4. Discussion

This descriptive study was aimed at exploring the attacking styles utilised in the 2021/22 WSL season and their relationship with the resulting shot on target. Specifically, 1179 attacking sequences (during 132 games) that led to shots on target were analysed, where most shots (27.23%) were originating from combinative organised attacks, while only 1.27% shots were from fast organised attacks. The sample also highlighted that 32.57% of those shots resulted in

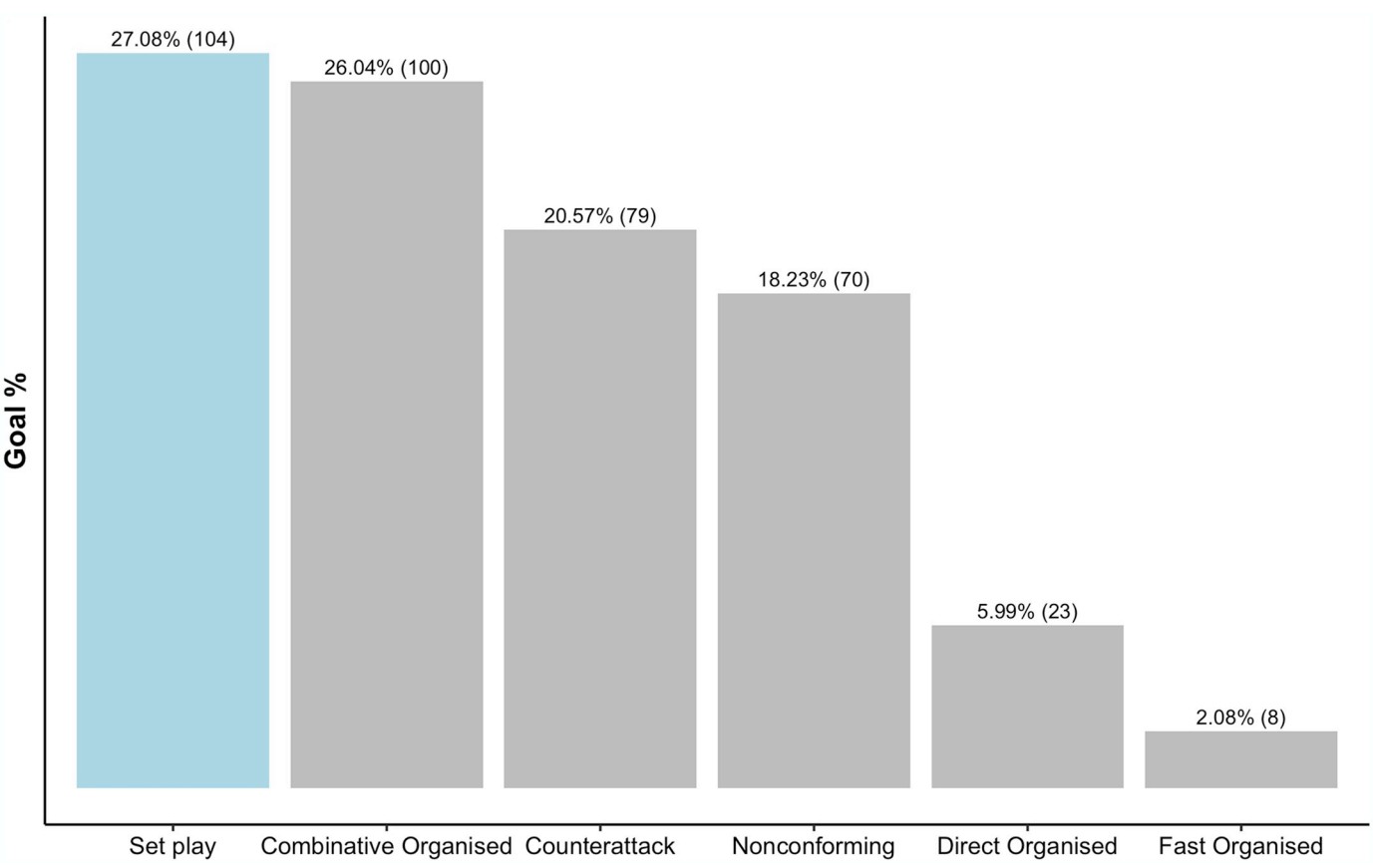

**Fig 4. Percentage of goals scored from the shots on target based on the different types of attacks in the 2021/2022 WSL season.**

goals, with most yielding from set plays (27.08%) and combinative organised attacks (26.04%). Interestingly, although only 2.08% of goals in the sample were from fast organised attacks, the results highlighted that 53.33% of the attacks originating from that style resulted in goals, with set plays also demonstrating more than 40% goal conversion rates. Importantly, the inferential analysis highlighted the prevalence of a significant relationship (p = 0.043) between style of attack and outcome of the shot on target in the 2021/22 WSL season, with set plays resulting in significantly (AR = 2.45) more goals than expected. Nevertheless, since the small effect size (V = 0.1) indicated that the observed association was weak and because the study design adopted is primarily descriptive, outcomes from high-quality observational studies may be necessary prior to making strong generalisable judgements from the results.

### 4.1. Attacking styles and shots on target

As specified in the introduction, since there is limited literature focused on examining the attacking styles in women's football, the remaining sections of this paper will rely on male focused research (although questions can be raised on the direct relevance) to compare the key outcomes formulated from this study. Thereby, it was firstly important to investigate the potential reasons for most shots on target in the 2021/22 WSL season examined in this study to be manifested from combinative organised attacks. Explicitly, as highlighted in Table 1 (see number of passes KPI), combinative organised attacks characteristically contain longer passing

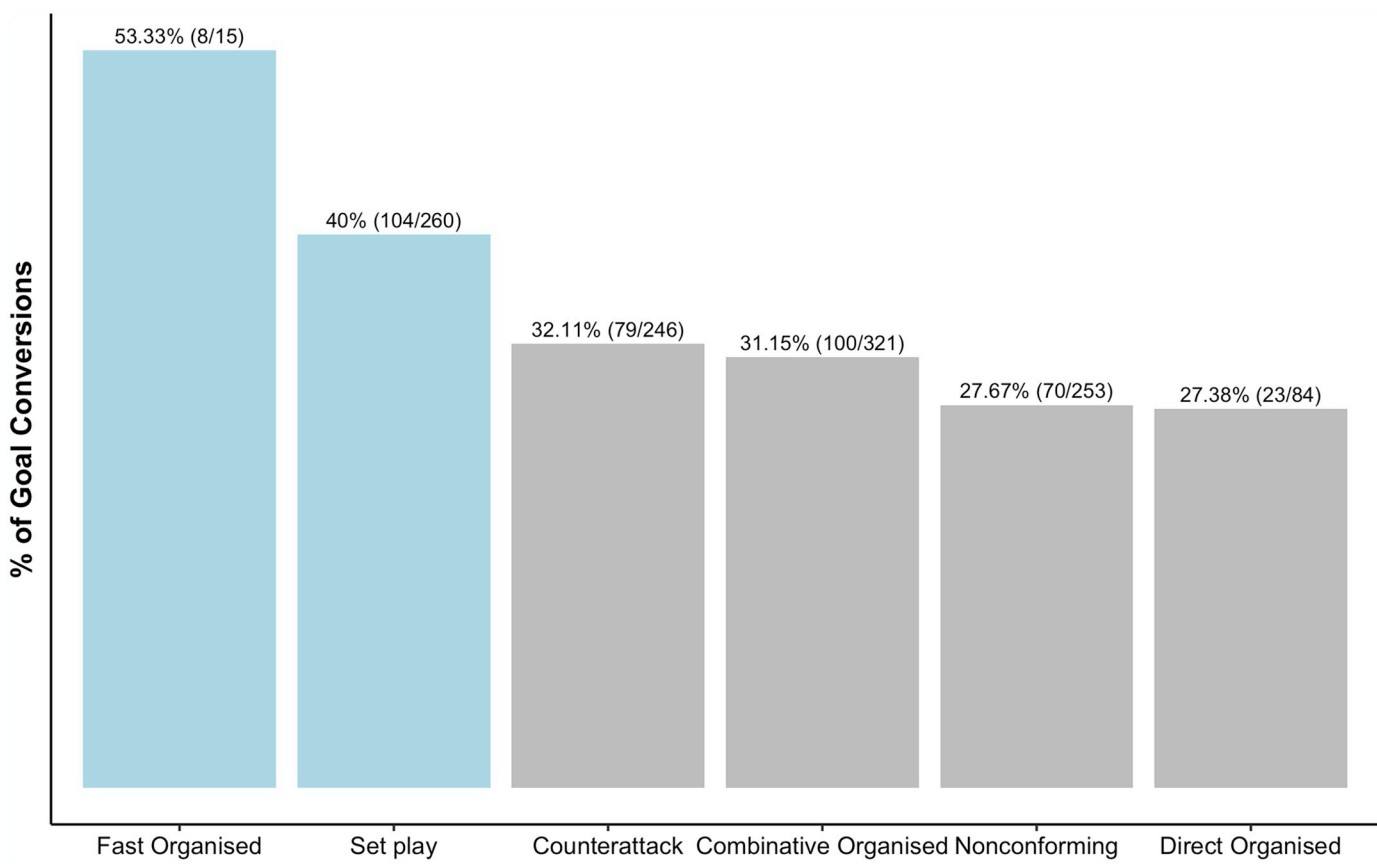

**Fig 5. Conversion rate of shots on target from the different types of attacks in the 2021/2022 WSL season.**

sequences prior to a shot occurring. While early football research concluded that short passing sequences resulted in a greater number of goals scored [55] such judgements have been contested with more contemporary studies finding attacking sequences consisting of three passes or more resulting in greater number of goals being scored [56]. Specifically, high number of consecutive passes has been associated with more effective penetration of space and exploitation of imbalances in the opponent's defence [57]. The constant movement of possession between players of the attacking team aims to force gaps within the opposition's defensive structure. Precise passes are then played to the player best positioned to execute a successful attempt on goal [20]. Players that attempt shots in more space and in closer proximity to the goal have a greater chance of the resulting shot being on target and achieving a goal [58]. Research has termed this area 'the scoring pentagon' outlining its importance in enhancing a team's shooting accuracy, conversion rate and overall goal-scoring proficiency [39]. Therefore, combinative organised attacks may characteristically cultivate a larger amount of those high percentage shots and could justify the high frequency of shots on target noted in the present study. It should be considered that the successful execution of longer, low risk passing sequences often heavily relies on the technical abilities of each player in this system [59]. More technically proficient players often play for higher ranked teams allowing for a more successful implementation of combinative organised attacks and could provide reason as to why those teams execute more shots on target, score more goals and win more games [60]. Hence, such judgements highlight that the current study can be extended to the micro-level, where

associations between the attacking styles based on team rankings in the WSL can be examined. However, Mitrotasios et al. [39] compared the implementation and effectiveness of different attacking styles, where the results suggested that combinative organised attacks are more frequently employed in male football compared to the women's game.

Conversely, the results highlighted that fast organised attacks contributed the lowest proportion of shots on target (1.27%) and goals (2.08%) in the considered sample. This result supports previous research, which found that fast organised attacks had resulted in less goals and goal-scoring opportunities [61]. Moreover, González-Rodenas et al. [62] found that despite fast attacks achieving more offensive penetration, they failed to create more goal scoring opportunities compared to combinative attacks. Perhaps, this is because the low number of shorter passes executed across the width of the pitch during a short period is likely to limit to progression of the ball towards the opponent's goal. Subsequently, enforcing shots to be taken further away from the opponent's goal, leading to more challenging positions to hit the target successfully [15]. Specifically, it is likely that those shots will be taken outside 'the scoring pentagon', resulting in reduced effectiveness of the shot. Thus, reducing the frequency of shots on target achieved and goals scored [39]. Nevertheless, although fast attacks may lead to limited instances of direct shots on target, they have been linked to the creation of successful attacking sequences like set plays (direct free-kicks, penalties and corner kicks) [30].

## 4.2. Goals from attacking styles

The present study highlights that set plays led to the most goals (27.08%), while also resulting in effective conversion rates (40%) in the 2021/22 WSL season. The goal-scoring effectiveness of set plays has been heavily acknowledged within research that had examined multiple competitions (in alignment with current results), where they appear to account for approximately 25–33% of goals scored [30, 63]. Importantly, the goal-scoring effectiveness from set plays have been examined within women's football [36, 43, 64, 65]. Specifically, Beare and Stone [64] summarised that corner kicks occur more frequently in women's football and were more efficient in producing goals compared to male football. However, research further indicates that less goals were produced from set plays, such as corners during international tournaments in women's football [65, 66]. The contrasting nature of the prior judgements may perhaps indicate the impact of physical differences of players in WSL teams to goals scored from set plays. Specifically, we hypothesise that WSL teams with greater representation of international players may pose a physical dominance during set plays. This possibly rationalises the large proportion of goals from set plays observed in the WSL season considered in this study. Conversely, this effect may be minimised during international tournaments due to uniform physical representation of players across national teams. Nevertheless, further research that examines the relationships between attacking styles, shot outcomes and the physical characterises of players in WSL teams (e.g., to examine set play performance of WSL teams on goal scoring opportunities in relation to team quality—i.e., composition of international versus non-international players) may be necessary to evaluate such prior hypothesis. Furthermore, research also indicates that set plays characteristically allow a greater number of attacking players to take a shot at goal, direct passes into 'the scoring pentagon', shots on a proximity to the goal and enforces less time pressure on the attacker to make an effective decision regarding a strategic attacking play—all of which may contribute to high goal conversion rates [35].

In terms of conversation rates, Hughes and Franks [24] noted that shorter, more direct attacks had higher shot conversion rates than longer sequences. Perhaps this justifies the outcome that fast organised attacks had the highest conversation (53.33%) rate among the attack styles. Fast organised attacks aim to use quick, vertical passes to exploit gaps and imbalances

within the opposition's defensive structure [23]. Explicitly, high percentage of penetrative passes, noted as extremely effective within research, enables the attacking team to move beyond the defensive unit and retain ball possession to have an attempt at the goal [28, 67]. Although this may occur infrequently during quick, wide attacks that contain shorter passes, if an opportunity arises, shots can be made within 'the scoring pentagon' [68]. Thereby, potentially maximising the conversion rate of the shots on target taken during those fast organised attacks. However, such judgements should be undertaken with care due to the low proportion of goals (2.08%) originating from fast organised attacks in the overall sample of shots considered in this study.

## 4.3. Relationship between attacking style and shot on target

From an inferential analysis perspective and as specified previously, the results indicated a significant association ($p = 0.043$) between the style of attack and the shot outcome within the 2021/22 WSL season. Interestingly, perhaps aligning to the above result and although the outcome of the shots were not considered, a study by González-Rodenas et al. [62] found significant differences between the attacking styles utilised to create goal scoring opportunities in male football. Specifically, in comparison to combinative attacks, counterattacks were more effective at creating scoring opportunities, while direct attacks were less likely to create chances and no differences were observed in relation to fast attacks. Thereby, when comparing such outcomes to the findings of the current study, from a purely statistical perspective and as discussed previously, because only shots originating from set plays led to significantly more goals than expected (AR = 2.45), further research outcomes based on high-quality observational designs is necessary (mainly because designing a randomised control trial for this if type of tactical research may not be practically viable) to examine justifications for the non-significant contributions of combinative organised (AR = -1.29), counterattack (AR = -0.69), direct organised (AR = -1.33) and fast organised (AR = 1.6) attacks to shot outcomes. Such judgements are further justified due to the small effect size (V = 0.1) of the chi-square test, where replication studies may be necessary to examine if the observed effect is consistent when examining attacking sequences in other WSL seasons. Furthermore, research heavily acknowledges the influence of contextual factors on the goal-scoring proficiency of different attack styles [34]. Hence, contextual dimensions like match location, opposition level, team level, match status, time of game, etc. associated with a team's tactical style, can influence performance trends, patterns and overall statistical significance, all of which are important to consider when generating specific practical conclusions [31, 32].

## 4.4. Study limitations

Beyond statistical considerations pertaining to small effect sizes of the chi-square test discussed previously, one of the key limitations of the current study was the categorisation and the subsequent exclusion of 'nonconforming attacks' from the final analysis. Specifically, 'nonconforming attacks' accounted for 21.46% of all attacking sequences and 18.23% of goals considered in the sample. this represents a considerable proportion of attacks and goals scored within the considered WSL season and therefore limits the formulation of strong generalisable conclusions from the results. Importantly, we didn't consider resolving this issue in this study since our objective was to provide an initial descriptive account of the attacking styles and their potential relationships to shot outcomes. However, we believe this situation creates an additional opportunity for future research to explore how different data analysis approaches like machine learning models can be used to categorise those attacks not meeting the criteria of the five styles considered in the current study. Specifically, if the objective is to

categorise the nonconforming attacks to the existing five styles, perhaps a supervised classification approach can be used [69]. Alternatively, an unsupervised classification technique like clustering may also be used to generate fresh clusters of attack styles based on the KPI and action variables specified in Table 1 [70]. However, such machine learning approaches may need to utilise techniques like 'one-hot encoding' to transform the categorical data of the KPIs conditions into numeric representations suitable for classification or clustering. Subsequently, we have provided the raw dataset of the current analysis to support such future research.

## 5. Conclusion

Overall, the outcomes of this study indicate a significant association between attacking styles and shot outcomes in the 2021/22 Women's Super League (WSL) season. From a coaching perspective, due to the highest prevalence of shots in the sample, if the offensive tactics in the WSL are focused on creating more shots on target, perhaps there is an opportunity to formulate attacking sequences based on combinative organised attacks. Nevertheless, despite generating the least shots on target and goals in this study, the high conversion rates of shots may indicate possibilities to adopt fast organised attacks in tactical scenarios aimed at maximising shooting opportunities. Specifically, while it may not lead to direct shooting chances, attacks originating from fast organised attacks may create opportunities for set plays. Thereby, potentially resulting in positive outcomes since the study outcomes indicate that significantly more goals than expected were scored from set plays in the 2021/22 WSL season. Finally, owing to the small effect size of the statistical test and because the adopted study design is mainly descriptive, future research should focus on implementing replication studies (perhaps considering different WSL seasons), alongside high-quality observational designs to further validate the current findings (e.g., observed associations). Additionally, there are also opportunities to investigate micro-level relationships between attacking strategies, shot outcomes and the physical characterises of players in WSL teams. Finally, future research can also explore the implementation of machine learning models to classify or cluster attacking strategies based on the data generated from the current study.

## Supporting information

**S1 Table. Supplementary information for this manuscript.**
(DOCX)

## Author Contributions

**Conceptualization:** Lizzie Craven, Jayamini Ranaweera.

**Data curation:** Lizzie Craven, Patrick Oxenham.

**Formal analysis:** Lizzie Craven, Jayamini Ranaweera.

**Investigation:** Lizzie Craven, Patrick Oxenham, Jayamini Ranaweera.

**Methodology:** Lizzie Craven, Patrick Oxenham, Jayamini Ranaweera.

**Project administration:** Lizzie Craven.

**Resources:** Lizzie Craven.

**Software:** Lizzie Craven.

**Supervision:** Patrick Oxenham, Jayamini Ranaweera.

**Validation:** Lizzie Craven, Patrick Oxenham, Jayamini Ranaweera.

**Visualization:** Lizzie Craven, Jayamini Ranaweera.

**Writing – original draft:** Lizzie Craven.

**Writing – review & editing:** Lizzie Craven, Patrick Oxenham, Jayamini Ranaweera.

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
