## [Decision Letter · Decision Letter 0]

11 Nov 2024

PONE-D-24-35271Analysis of attacking styles and goal-scoring in the 2021/22 Women’s Super League.PLOS ONE

Dear Dr. Ranaweera,

Thank you for submitting your manuscript to PLOS ONE. After careful consideration, we feel that it has merit but does not fully meet PLOS ONE’s publication criteria as it currently stands. Therefore, we invite you to submit a revised version of the manuscript that addresses the points raised during the review process.

We look forward to receiving your revised manuscript.

Kind regards,

Anwar P.P. Abdul Majeed

Academic Editor

PLOS ONE

Journal Requirements:

Reviewers' comments:

Reviewer's Responses to Questions

**Comments to the Author**

1. Is the manuscript technically sound, and do the data support the conclusions?

Reviewer #1: Yes

2. Has the statistical analysis been performed appropriately and rigorously? 

Reviewer #1: Yes

3. Have the authors made all data underlying the findings in their manuscript fully available?

Reviewer #1: Yes

4. Is the manuscript presented in an intelligible fashion and written in standard English?

Reviewer #1: Yes

5. Review Comments to the Author

Reviewer #1: This study aimed to describe goal scoring in relation to different attacking styles during a Women’s Super League (WSL) season. This is a well written paper and there is limited research in women’s football. The authors should be commended for this novel research.

Lines 41-43: Insert references to show that your argument is based on empirical evidence.

Line 105: If you say previous research, insert references.

Although the authors intent to close the gap in the literature, but it is important to provide a strong justification of the study.

Consider rephrasing the statement below: “Finally, once the identification KPIs and operational definitions was completed and their validity was established within the literature, they were presented to a panel of experts who further asserted the validity and relevance of the KPI, action variables and categorisation of the attack styles from them.” Make sure that you proofread your paper. Further, some words are capitalised unnecessary (e.g., Set Play). Also, be clear and concise.

Line 166: I think your study is observational.

Line 206: Consider removing this sentence, “Afterwards, bar charts were used to visualise the descriptive results.”

Lines 268-281: There are no references to support or contrast your findings.

6. PLOS authors have the option to publish the peer review history of their article (what does this mean?). If published, this will include your full peer review and any attached files.

Reviewer #1: No

---

## [Author Response · Author response to Decision Letter 0]

27 Nov 2024

Dear Editor and Reviewer,

On behalf of all the authors, we are extremely grateful for your time and efforts in reviewing our manuscript and providing suggestions to improve it. Subsequently, we considered all comments and amended the manuscript accordingly. Moreover, we have provided responses to each comment within the 'Responses to Reviewers' documents attached with the submission. 

Many thanks

The authors

---

## [Decision Letter · Decision Letter 1]

24 Jan 2025

Analysis of attacking styles and goal-scoring in the 2021/22 Women’s Super League.

PONE-D-24-35271R1

Dear Dr. Ranaweera,

We’re pleased to inform you that your manuscript has been judged scientifically suitable for publication and will be formally accepted for publication once it meets all outstanding technical requirements.

Kind regards,

Anwar P.P. Abdul Majeed

Academic Editor

PLOS ONE

Additional Editor Comments (optional):

Reviewers' comments:

Reviewer's Responses to Questions

**Comments to the Author**

1. If the authors have adequately addressed your comments raised in a previous round of review and you feel that this manuscript is now acceptable for publication, you may indicate that here to bypass the “Comments to the Author” section, enter your conflict of interest statement in the “Confidential to Editor” section, and submit your "Accept" recommendation.

Reviewer #1: All comments have been addressed

2. Is the manuscript technically sound, and do the data support the conclusions?

Reviewer #1: Yes

3. Has the statistical analysis been performed appropriately and rigorously? 

Reviewer #1: Yes

4. Have the authors made all data underlying the findings in their manuscript fully available?

Reviewer #1: Yes

5. Is the manuscript presented in an intelligible fashion and written in standard English?

Reviewer #1: Yes

6. Review Comments to the Author

Reviewer #1: The paper is well written. There is little information about women's football worldwide. The authors have addressed all the comments and the paper should be accepted..

7. PLOS authors have the option to publish the peer review history of their article (what does this mean?). If published, this will include your full peer review and any attached files.

Reviewer #1: **Yes: **Alliance Kubayi

---

## [Editor Report · Acceptance letter]

28 Jan 2025

PONE-D-24-35271R1 

PLOS ONE

Dear Dr. Ranaweera, 

I'm pleased to inform you that your manuscript has been deemed suitable for publication in PLOS ONE. Congratulations! Your manuscript is now being handed over to our production team.

Kind regards, 

on behalf of

Dr. Anwar P.P. Abdul Majeed 

Academic Editor

PLOS ONE